# Advances in Chromatographic Analysis of Phenolic Phytochemicals in Foods: Bridging Gaps and Exploring New Horizons

**DOI:** 10.3390/foods13142268

**Published:** 2024-07-18

**Authors:** Jorge Antonio Custodio-Mendoza, Patryk Pokorski, Havva Aktaş, Alicja Napiórkowska, Marcin Andrzej Kurek

**Affiliations:** Department of Technique and Food Development, Institute of Human Nutrition Sciences, Warsaw University of Life Sciences (WULS-SGGW), 02-776 Warsaw, Poland; jorge_custodio-mendoza@sggw.edu.pl (J.A.C.-M.); patryk_pokorski@sggw.edu.pl (P.P.); havva_aktas@sggw.edu.pl (H.A.); alicja_napiorkowska@sggw.edu.pl (A.N.)

**Keywords:** analytical chemistry, anthocyanins, catechins, CE, chromatography, flavonoids, food analysis, GC, HPLC, LCxLC, MS/MS, nano-LC, phenolic acids, phytochemicals, SFC, UV

## Abstract

Chromatographic analysis of phenolic phytochemicals in foods has significantly advanced over the past decade (2014–2024), meeting increasing demands for precision and efficiency. This review covers both conventional and advanced chromatographic techniques used for detecting phenolic phytochemicals in foods. Conventional methods like High-Performance Liquid Chromatography, Ultra High-Performance Liquid Chromatography, Thin-Layer Chromatography, and Gas Chromatography are discussed, along with their benefits and limitations. Advanced techniques, including Hydrophilic Interaction Liquid Chromatography, Nano-LC, Multidimensional Liquid Chromatography, and Capillary Electrophoresis, are highlighted for their innovations and improved capabilities. The review addresses challenges in current chromatographic methods, emphasizing the need for standardized and validated procedures according to the Food and Drug Administration, European Cooperation for Accreditation of Laboratories, and The International Organization for Standardization guidelines to ensure reliable and reproducible results. It also considers novel strategies for reducing the environmental impact of chromatographic methods, advocating for sustainable practices in analytical chemistry.

## 1. Introduction

Phenolic phytochemicals, the powerhouse bioactive components found in a diverse range of foods, particularly fruits, vegetables, and whole grains, are of paramount importance in food science and health [1]. Renowned for their robust antioxidant, anti-inflammatory, and anticancer properties, these compounds not only enrich the nutritional value of our diets but also play a pivotal role in our overall health [1,2,3]. The meticulous analysis of these phenolic compounds is therefore not just a scientific pursuit, but a critical stride towards comprehending and harnessing their potential health impacts [3]. 

Phenolic phytochemicals are organic compounds with phenolic hydroxyl groups in their structures. They are found in most plants and contribute to the bitterness and color of a variety of foods [4]. These phytochemicals include flavonoids, flavones, flavanonols, flavanones, anthocyanidins, isoflavones, lignans, stilbenes, curcuminoids, phenolic acids, and tannins (Figure 1). 

Recent advances in chromatographic technologies have enhanced these bioactive compounds’ detection, identification, and quantification. Techniques such as Ultra High-Performance Liquid Chromatography (HPLC) coupled with advanced detectors like Quadrupole Time-of-Flight Mass Spectrometry (Q-TOF-MS) have provided detailed and accurate phenolic profiles. This methodological evolution offers a more nuanced understanding of phenolic structures and their bioactivities, which is crucial for linking dietary intake to potential health outcomes [5]. 

In addition to UHPLC/Q-TOF-MS, recent innovations in biosensing technologies have introduced carbonaceous nanomaterials-based sensors, which offer rapid, sensitive, and reliable phenolic detection. These biosensors are particularly promising for applications requiring real-time monitoring and environmental analysis, reflecting a trend toward integrating more dynamic and versatile analytical tools in phenolic research [6].

Significant strides in structural elucidation techniques have also complemented the chromatographic analysis of phenolic compounds. Modern chromatographic setups, often coupled with sophisticated spectroscopic methods like Nuclear Magnetic Resonance (NMR) and Mass Spectrometry (MS), allow for the identification and detailed structural analysis of complex phenolic molecules [7]. This level of detail is critical for exploring the specific interactions of phenolics with biological targets, which can inform the therapeutic and nutraceutical uses of these compounds [7,8]. 

This review will cover these advancements and discuss the implications of enhanced chromatographic techniques in analyzing food phenolics. The established and widely used methods are covered under conventional chromatographic techniques, while newer and more specialized liquid chromatography-based methods developed to address specific challenges are included in the advanced techniques section. 

By highlighting the technological progress and its applications, the review will shed light on how these developments have influenced academic research and practical applications in food science and nutrition. The integration of advanced chromatographic techniques is pivotal for advancing our understanding of phenolic compounds’ complex role in health and disease, paving the way for future innovations in food technology and preventive medicine.

## 2. Conventional Chromatographic Techniques for Phenolic Phytochemicals

Chromatographic techniques are commonly used to analyze phenolic phytochemicals, with High-Performance Liquid Chromatography and UV detection (HPLC-UV) being one of the most prominent. This method is widely used due to its robustness, well-established analytical methods, and extensive spectrum libraries that aid in identifying phenolic structures [9]. The process involves extracting and filtering a food sample with a solvent and then injecting the prepared sample into the HPLC apparatus [10,11]. Under high pressure, the sample passes through a column filled with a stationary phase, where phenolic chemicals are separated based on their chemical affinities to the adsorbent material. A UV detector then measures the light absorbance of the eluted compounds at specific wavelengths, allowing for both qualitative and quantitative analysis [12].

Figure 2 presents the number of articles published using chromatographic techniques according to the Web of Science using “phenolic”, “phytochemicals”, “chromatography”, “HPLC”, “UHPLC”, “GC”, “TLC”, “Nano-LC”, “SFC”, “CE”, and “LCxLC” as keywords (https://www.webofscience.com/, accessed on 31 June 2024). 

Gas Chromatography (GC) is another conventional method for detecting volatile phenolic chemicals. GC is beneficial for compounds that can be vaporized without decomposition [13]. This method distinguishes between volatile compounds based on their boiling points and their interaction with the column’s stationary phase. When further structural details are required, the compounds are typically identified using a Flame Ionization Detector (FID) or MS. While GC is renowned for its high resolution and sensitivity, it is limited to analyzing volatile substances or those that can be converted into volatile derivatives, which excludes the analysis of non-volatile phenolics [14,15]. 

Thin-layer chromatography (TLC), in both conventional and high-performance (HPTLC) formats, is regarded as a flexible and high-throughput liquid chromatography technology with several significant applications [16]. This technique separates materials by interacting with a thin layer of adsorbent connected to a plate with low molecular weight molecules [16]. A variety of adsorbents are used to separate various substances [17]. TLC has historically played a role in the analysis of phenolic compounds [16,17]. This method involves using high-quality filter paper as the stationary phase and a suitable solvent as the mobile phase [16,17]. The principle of TLC is based on the differential partitioning of compounds between the stationary and the mobile phases, which allows for the separation of components based on their solubilities and interaction with the paper medium [16,17]. The solvent travels ahead by capillary action, bringing soluble molecules with it. Low porosity paper causes the solvent to travel slowly, but thicker paper increases sample capacity [17].

While conventional chromatographic methods have been instrumental in advancing our understanding of phenolic compounds in foods, they also face several limitations and challenges. These issues can impact these techniques’ efficacy, accuracy, and applicability in contemporary scientific research and industrial applications. One of the primary challenges associated with conventional chromatographic techniques is the limitation in resolution and sensitivity. When dealing with complex food matrices, including closely related phenolic chemicals, HPLC with UV detection may lack resolution, resulting in co-elution or overlapping peaks [18]. This restriction hinders the identification and measurement of these substances. Similarly, the sensitivity of UV detection may be insufficient to identify phenolic chemicals in low quantities, potentially resulting in an underestimation of their amounts in samples [19,20]. 

While GC has excellent resolution and sensitivity for volatile chemicals, it only applies to analytes that can be vaporized without decomposition. Non-volatile phenolics must be chemically modified (derivatized) to make them suitable for GC analysis, which might change their natural state and potentially alter findings [21]. TLC has even lower resolution and sensitivity, making them best suited for qualitative screens rather than extensive quantitative analysis [22]. Extensive sample preparation is another critical problem with these techniques. Sample preparation in HPLC and GC may be complex and time-consuming, requiring many processes such as extraction, purification, and, in some instances, derivatization [23]. This increases the danger of compound degradation or loss and reduces the throughput of these procedures, making them unsuitable for high-throughput investigation. The labor-intensive nature of these preparations raises the possibility of human error, lowering the repeatability and dependability of the results [24]. 

Conventional chromatographic procedures can involve massive quantities of organic solvents, which are expensive and hazardous to the environment. Disposing of toxic solvents affects the environment, making conventional methods less sustainable than more recent, eco-friendly procedures that employ less hazardous solvents or solventless systems [25].

To summarize, while conventional chromatographic techniques have offered fundamental insights into phenolic chemicals in foods, their limits and constraints need continued developments in chromatographic technology. These advancements are critical for obtaining improved resolution, enhanced sensitivity, more efficient sample handling, and more environmental sustainability in the analysis of phenolic chemicals. 

## 3. Advancements in Liquid Chromatography-Based Methods for Polyphenol Phytochemical Analysis of Foods

Liquid Chromatography-based methods take center stage in the analysis and determination of polyphenol phytochemicals due to their compatibility with aqueous extracts, which, in most cases, simplify extraction and sample preparation [14,17]. Recent advancements encompass hyphenated techniques as in Liquid Chromatography–Mass Spectrometry (LC-MS) with the use of specific functionalized stationary phases as in Hydrophilic Interaction Liquid Chromatography (HILIC), offering enhanced sensitivity, specificity, and structural elucidation capabilities [24,25]. Capillary Electrophoresis (CE) and Capillary Electrochromatography (CEC) are also explored for their contributions to the high-efficiency separation of phenolic compounds. Through these innovative approaches, researchers are better equipped to unravel the complex profiles of polyphenols in various food matrices, facilitating a deeper understanding of their nutritional and functional properties [17,20,24,25].

### 3.1. Hydrophilic Interaction Liquid Chromatography 

HILIC is a promising technique for separating polar carbohydrates and semi-polar aromatic compounds. Unlike reverse-phase liquid chromatography (RP-LC), HILIC utilizes a polar stationary phase, enhancing the retention and separation of hydrophilic compounds [25,26]. HILIC has gained traction as a valuable technique for separating polar compounds, particularly those with high polarity, such as carbohydrates and amino acids. HILIC is versatile and compatible with various analytes, including small polar molecules, peptides, proteins, and bioactive compounds [27]. HILIC has been used in the analysis and determination of a broad spectrum of bioactive compounds such as polyphenols, amino acids, alkaloids, nucleosides, nucleotides, vitamins, and phytochemicals, including flavonoids, phenolic acids, catechins, anthocyanins, flavonols, and procyanidins in different foodstuff, including fruits, vegetables, teas, vines, and dietary supplements, among others [28,29,30,31,32,33,34,35,36,37,38,39,40,41]. 

Table 1 presents different analytical methods employing HILIC for the determination of polyphenol phytochemicals in various food samples, along with their respective instrumental conditions and detection systems.

Different column dimensions (100–250 × 1.0–4.6 mm) and particle sizes (1.3–5 µm) are employed in this method along with various combinations of water and organic solvents (acetonitrile, methanol) to achieve optimal separation in the analysis of different food samples (Table 1). Remarkably, chemical modifiers such as weak organic acids (e.g., acetic, formic, trifluoroacetic acids) and organic salts (e.g., ammonium formate, sodium acetate) are utilized to improve chromatographic resolution and analyte separation [35,37,38,40]. One key advantage of HILIC is its compatibility with different detection systems. MS, UV-vis spectroscopy (UV-vis), diode array detection (DAD), evaporative light scattering detection (ELSD), and tandem MS (MS/MS) are employed for the quantification and identification of analytes [25,30,34,37,38,39,41]. 

Additionally, MS-based detection offers high sensitivity and specificity, allowing for the structural elucidation of analytes [29,36,38,40]. At the same time, UV-vis and DAD provide quantification based on absorbance at specific wavelengths, for which the use of higher concentrations of acids is required (1–10%) depending on the chemical nature of the analytes [41]. Moreover, the total analysis time varies among the methods, ranging from a few minutes to several tens of minutes. Factors influencing the analysis time include the complexity of the sample matrix, the number of analytes targeted, the chromatographic conditions, and the detection system used [25,26,27,28,29,30,31,32,33,34,35,36,37,38,39,40,41].

### 3.2. Nano-Liquid Chromatography 

Nano-LC is a powerful tool for determining phenolic phytochemicals. It requires reduced sample sizes and mobile phase volumes. The miniaturization of Nano-LC leads to a more environmentally friendly and cost-effective process, which perfectly aligns with current trends in *green analytical chemistry* [42]. 

Nano-LC has been used to determine phenolic acids, flavonoids, alkaloids, and other phenolic compounds in various foods, including olive oil, tea, citrus juices, and dietary supplements [43,44,45,46,47,48,49]. These applications typically use reverse-phase C18 stationary phases with different particle sizes, like those in RP-LC. The chromatographic columns have smaller inner diameters (0.075–0.1 mm) and are either purchased from suppliers [49] or prepared in the lab. Lab preparation involves removing the stationary phase from commercial chromatographic columns, dissolving it in pure solvents (commonly acetone), and introducing the phase dispersion into capillary tubes by immersion [43,44,45,46,47,48]. After homogenization, the columns are conditioned by passing several column volumes of organic solvents recommended by the manufacturer and then cut to a usable length of 10–15 cm [43,44,45,46,47,48]. 

Table 2 presents different Nano-LC methods for determining phenolic phytochemicals in various food samples and their respective instrumental conditions and detection systems.

Compared to conventional LC methods, Nano-LC uses 100–300 times smaller mobile phase volumes, although it employs similar mobile phase compositions to analyze phenolic phytochemicals. Flow rates range from 350 to 1200 nL/min, depending on the method and sample analyzed. Detection systems include UV-MS, MS, UV-vis, and Q-TOF-MS, providing distinct levels of sensitivity and specificity for analyte detection. Although Nano-LC enhances the separation of phenolic compounds with superior resolution and lower detection limits compared to conventional LC methods, the in-lab preparation of Nano-LC columns poses significant challenges to feasibility and replicability due to the technical complexity, precision required, specialized equipment and materials needed, time-consuming processes, reproducibility issues, and high costs. These factors make it difficult to consistently produce high-quality columns, impacting Nano-LC methods’ reliability and widespread application.

### 3.3. Supercritical Fluid Chromatography

Supercritical Fluid Chromatography (SFC) is an innovative technique that utilizes supercritical fluids, typically CO₂, as the mobile phase. [50,51]. Table 3 presents different analytical methods that use SFC to determine phenolic phytochemicals in various food samples and their respective instrumental conditions and detection systems. CO_2_ is frequently utilized as a supercritical fluid due to its low critical temperature (31 °C) and critical pressure (7.3 MPa), making it compatible with commercially available equipment. In SFC, the mobile phases consist of either CO_2_ alone or CO_2_ mixed with organic modifiers such as methanol (MeOH), ethanol (EtOH), or a combination of both, with flow rates varying between 10 and 22.5 mL/min based on the analyte’s chemistry and sample complexity. The stationary phases employed are akin to those utilized in HPLC and UHPLC, encompassing HSS SB C18, Cortecs C18, Chiralpak AD, Diol, and BEH-2EP columns, with particle sizes ranging from 1.7 μm to 10 μm [52,53,54,55,56,57].

SFC presents several advantages, including high efficiency, rapid separation times, and reduced environmental impact due to non-toxic and non-flammable supercritical fluids. It aligns well with green analytical chemistry principles [42] by minimizing the utilization of hazardous substances and waste. SFC reduces solvent waste, operates with greater energy efficiency, and employs safer solvents than conventional liquid chromatography methods [50,51]. Moreover, SFC supports real-time analysis for pollution prevention, enhances safety for laboratory personnel, and maintains high efficiency and performance in separations [50,51].

SFC proves particularly effective in separating phenolic isomers, which can be challenging with conventional LC methods, and is compatible with various detection systems, including UV-vis (PDA, DAD), MS, and MS/MS). The unique properties of supercritical fluids, such as low viscosity and high diffusivity, enhance separation efficiency and resolution. SFC’s compatibility with various detectors, including MS, further extends its application in the analysis of phenolic compounds.

### 3.4. Multi-Dimensional Chromatography

Multi-Dimensional Chromatography (MDC) is an advanced technique that integrates multiple chromatographic methods to comprehensively separate complex mixtures. This approach is particularly beneficial for profiling complex phenolic mixtures, which often contain compounds with similar structures and retention times [58,59]. By utilizing two or more different chromatographic systems in tandem, MDC allows for separating phenolics that might co-elute in a single-dimensional system [58]. 

In Comprehensive Two-Dimensional Liquid Chromatography (LCxLC), two chromatographic separation mechanisms are combined sequentially, typically with different selectivities [58,59,60]. The first dimension separates the sample mixture using one chromatographic method, and the eluate is then fractionated onto a second column in the second dimension, offering enhanced peak capacity and resolution [58,59,60,61]. This approach allows for better separation of closely eluting compounds and improved detection sensitivity, making it particularly useful for comprehensively analyzing complex mixtures like phenolic phytochemicals in foods. For instance, coupling RP-LC with HILIC can enhance the separation of phenolics based on their different chemical properties. LCxLC improves the resolution and peak capacity and provides a more detailed and comprehensive analysis of phenolic profiles, facilitating the identification and quantification of minor and co-eluting compounds [60,61]. 

The analysis of flavonoids, alkaloids, phenolic acids, cannabinoids, procyanidins, organic acids, lignans, catechins, terpenes, and anthocyanins in different foods, including safflower, *Rhus coriaria, cannabis, Gelsemium elegans, Cuscuta Chinensis, Ceylon tea, Cannabis sativa*, various berries, grape-related products, green tea, apples, and Uncaria sessilifructus has been reported using LCxLC with either RP-LC and HILIC columns or their combination [62,63,64,65,66,67,68,69,70,71,72,73]. 

Stationary phases employed for these analyses include columns such as amide, C18, PFP, HILIC, SB-AQ, RP, CN, Diol, and SCX, with particle sizes ranging from 1.7 µm to 5 µm designed for high-efficiency separations [62,63,64,65,66,67,68,69,70,71,72,73]. Mobile phases typically comprise water and organic solvents like acetonitrile (ACN) or methanol, often augmented with additives such as formic acid (FA), ammonium formate (AF), or trifluoroacetic acid (TFA) to enhance separation and detection [62,63,64,65,66,67,68,69,70,71,72,73]. Flow rates vary significantly, from 0.01 mL/min to 3 mL/min, depending on the column and desired resolution [62,63,64,65,66,67,68,69,70,71,72,73]. 

Detection systems used include DAD, high-resolution mass spectrometry (HRMS), quadrupole time-of-flight mass spectrometry (Q-TOF-MS), and photodiode array (PDA) systems, offering high sensitivity and specificity for a wide range of compounds [62,63,64,65,66,67,68,69,70,71,72,73]. Total analysis times range from 50 to 170 min, balancing thorough separation with efficiency [62,63,64,65,66,67,68,69,70,71,72,73]. The methods demonstrate considerable variability in stationary and mobile phase compositions tailored to specific analytes and sample matrices.

Table 4 presents different MDC methods to determine food phenolic phytochemicals, along with their respective instrumental conditions and detection systems [62,63,64,65,66,67,68,69,70,71,72,73]. 

High efficiency is achieved using columns with smaller particle sizes and advanced materials, such as C18, amide, and HILIC [62,63,64,65,66,67,68,69,70,71,72,73]. The flexibility in detection systems allows for versatile applications depending on the target analytes, ranging from simple UV detection to advanced mass spectrometry.

These methods offer high analytical performance but have room for improvement when it comes to environmental friendliness and alignment with green analytical chemistry principles [42]. The prevalent use of organic solvents like acetonitrile and methanol, along with additives such as formic acid (FA) and trifluoroacetic acid (TFA), poses environmental concerns due to their toxicity and disposal challenges. While low flow rates and high-efficiency columns with smaller particle sizes contribute to reduced solvent consumption and shorter analysis times, some methods still have high solvent usage and longer analysis times, increasing their environmental impact. Although extremely sensitive, advanced detection systems like HRMS and Q-TOF-MS are energy-intensive. To better align with green analytical chemistry trends, these methods could incorporate more water-based mobile phases, reduce the reliance on toxic solvents, adopt micro and Nano-LC techniques, and enhance energy efficiency. While effective, these methods could benefit from modifications to become more sustainable and environmentally friendly.

### 3.5. Capillary Electrophoresis 

Liquid Chromatography (LC) and Capillary Electrophoresis (CE) share their fundamental principles and operational parameters for separating analytes using stationary and mobile phases. LC employs a packed column, while CE uses a capillary tube. Both methods use silica-based particles as the stationary phase and organic solvents or acids mixed with water-based buffers as the mobile phase. However, LC relies on differential partitioning for separation. In contrast, CE uses an electric field, electroosmotic flow, and electromigration, leading to faster separation times than UHPLC [74].

CE is particularly advantageous for analyzing phenolic compounds in complex food matrices, offering rapid and efficient separations with high resolution and sensitivity. This makes it ideal for determining bioactive compounds like polyphenols. Studies using CE with UV detection have successfully analyzed phytochemicals in various plant materials, such as chamomile flowers, Salvia species, edible flowers, and carob syrup. Researchers optimize conditions like capillary length, voltage, pH, buffer composition, and concentration to achieve good separation and quantification of phenolic compounds. [75,76,77,78,79] 

All methods use capillaries with an internal diameter of 50 µm, with lengths ranging from 55 cm to 100 cm [74,75,76,77,78,79,80,81]. Separation media are primarily borate buffers, using sodium tetraborate, with concentrations from 0.012 M to 1 M, tailored to the specific analytes and sample complexities. Most samples use Capillary Zone Electrophoresis (CZE), which separates analytes based on their charge-to-mass ratio, while Liu et al. (2023) used Pressurized CE (pCEC) for the simultaneous determination of 11 phenols in the famous conventional Chinese medicine Shihu, achieving higher resolution [77]. 

Moreover, Capillary Electrophoresis-Mass Spectrometry (CE-MS) combines CE’s high separation efficiency with mass spectrometry’s analytical capabilities [74,82]. CE-MS enables sensitive and specific detection and identification of phenolic compounds, making it suitable for analyzing complex phenolic mixtures and high-throughput applications [74,82]. The integration of CE with MS enhances the detection of a wide range of phenolics, including those at low concentrations, and provides detailed structural information for compound identification [74,82]. Analysis times range from 10 min for more straightforward samples like Salvia to 40 min for complex matrices like soybeans and berries.

Overall, CE methods (Table 5) are versatile for analyzing phenolic compounds in diverse food samples, offering high efficiency and alignment with green analytical chemistry principles using smaller amounts of solvents and reagents than conventional chromatography [42]. The choice of method depends on the specific requirements for sensitivity, resolution, and analysis time.

## 4. Advancement, Challenges, and Future Directions in Chromatographic Analysis of Phenolic Phytochemicals

### 4.1. Advancements and Limitations 

Chromatographic techniques such as HPLC, GC, TLC, and their advanced forms like HILIC, Nano-LC, SFC, and MDC provide robust tools for analyzing phenolic compounds (Figure 3). HPLC with UV detection is widely used due to its robustness, established methods, and extensive spectrum libraries, facilitating accurate identification and quantification of phenolics. GC is highly effective for volatile phenolics, offering excellent resolution and sensitivity, especially with detectors like FFID or MS. HILIC is beneficial for separating polar compounds, and SFC aligns with green chemistry principles by using CO₂ as a non-toxic mobile phase. Nano-LC and MDC provide high resolution and sensitivity, with MDC allowing comprehensive profiling of complex mixtures through multi-dimensional separation techniques.

Many factors can influence the performance of chromatographic methods, making it essential to optimize them for the best possible analyte separation. 

Column Dimensions and Particle Size: In LC-based techniques (HPLC, UHPLC, Nano-LC), the relationship between column length, diameter, and particle size is crucial [7]. Longer columns improve resolution but increase analysis time and backpressure [20]. Smaller diameters reduce band broadening and provide sharper peaks, while smaller particle sizes enhance separation but generate higher backpressure, requiring robust systems [7,21].Flow Rate: Properly optimizing the flow rate is vital. A high flow rate can lead to poor separation, while a low flow rate can cause excessive diffusion. An optimal flow rate balances efficiency and analysis time [73].Mobile Phase Composition and pH: The choice of solvents and their ratios affect the interaction of analytes with the stationary phase [7]. Acidified mobile phases are preferred due to their ability to enhance the resolution and peak shape of analytes. However, it is crucial to ensure that the use of acidic conditions is compatible with the specifications and limitations of the chosen detectors to avoid any potential damage or interference with detection sensitivity [7,21,30].Pressure: In UHPLC, higher pressure allows the use of smaller particle sizes, improving resolution and enabling faster flow rates without sacrificing performance, provided the system can handle the increased pressure [37].Injection Volume: Smaller injection volumes minimize band broadening, leading to sharper peaks and better resolution. Overloading the column can cause peak distortion [7,21,43,44].Stationary Phase Composition: The material and chemical properties of the stationary phase directly influence retention and separation. Reverse phase (non-polar) and HILIC (polar) phases are commonly used, with LCxLC combining both for complex separations [63,69].Temperature: Temperature is crucial in GC for separation efficiency [15]. In LC-based techniques, higher temperatures reduce viscosity and retention times, improving resolution, although it must be optimized to prevent degradation [7,21].

Conventional chromatographic methods face several limitations. HPLC with UV detection may struggle with resolution and sensitivity, particularly with complex food matrices potentially leading to co-elution and underestimation. GC is limited to volatile compounds, requiring derivatization for non-volatile phenolics, which can alter their natural states [15]. TLC and paper chromatography offer lower resolution and sensitivity, making them more suitable for qualitative rather than quantitative analysis [16]. Extensive sample preparation, involving extraction, purification, and derivatization, can be time-consuming and prone to errors, reducing throughput and repeatability [59]. Additionally, these techniques often use enormous amounts of organic solvents, posing environmental and sustainability challenges [7,21]. 

Despite their efficacy, conventional methods need further development to address these limitations. Advancements in chromatographic technology are crucial for achieving better resolution, sensitivity, and efficient sample handling. Research should focus on enhancing the environmental sustainability of these techniques by reducing solvent usage and adopting greener alternatives. Innovative methods like HILIC and Nano-LC show promise, but their technical complexity and high costs challenge widespread application [41,46,47]. Additionally, there is a need for more comprehensive studies on the applicability of these advanced techniques to a broader range of food matrices and phenolic compounds. Research into combining multiple chromatographic methods, as seen in MDC, could offer enhanced separation and detection capabilities, paving the way for more detailed and accurate phenolic profiling [71,72,73].

### 4.2. Ensuring Accuracy and Reliability: The Importance of Standardization and Validation 

Standardization of chromatographic methods is crucial for ensuring the accuracy and precision of analyses in the study of phenolic phytochemicals. By establishing consistent protocols for sample preparation, chromatographic conditions (such as column type, mobile phase composition, and flow rates), and detection methods, standardization minimizes systematic errors. It enhances reproducibility across different laboratories and studies [83,84,85] compounds but also improves the precision of results by reducing variability within replicate analyses. Moreover, standardized methods enable reliable data comparisons between different studies and laboratories, supporting robust scientific conclusions and facilitating the validation of findings [86,87]. Quality control measures embedded in standardized protocols, such as calibration curves and system suitability tests, further ensure that chromatographic systems operate within defined performance criteria, enhancing the reliability and credibility of analytical results [87]. Adherence to standardized chromatographic methods advances scientific research and meets regulatory requirements in various industries, safeguarding product quality and consumer safety [88,89].

Analytical method validation is critical for ensuring the accuracy and reliability of analytical results but comes with several challenges [90,91,92]. One major challenge is the complexity of food product matrices [90]. These matrices often contain compounds that can interfere with the analyte of interest, making it difficult to achieve accurate measurements [90,91]. Another challenge is consistently achieving precision and accuracy across different batches, operators, and instruments [90,91,92]. Variations in sample preparation, equipment performance, and environmental conditions can influence the reproducibility of results, requiring careful control and validation [90,91,92].

Additionally, determining the limits of detection (LOD) and quantification (LOQ) can be challenging, especially for analytes present in low concentrations (trace analysis) [93]. Methods must be sensitive enough to detect trace amounts reliably without being overly influenced by minor variations in sample conditions [90,91,92,93]. Ensuring specificity and selectivity is another hurdle, as methods must distinguish between the analyte of interest and structurally similar compounds or matrix components. Cross-reactivity and interference can compromise the accuracy of quantitative measurements, necessitating thorough validation of method performance [92].

Robustness ensures that methods remain stable and consistently perform under varying experimental conditions such as pH, temperature, and mobile phase composition [90,91,92,93]. Method transferability across different laboratories or instruments adds complexity, necessitating validation to ensure consistent performance in diverse settings [94]. Documenting validation studies and adhering to regulatory guidelines, though essential, can be time-consuming and resource-intensive.

International organizations such as the Food and Drug Administration (FDA), the European Cooperation for Accreditation of Laboratories (EURACHEM), and the International Organization for Standardization (ISO 17025) provide validation guidelines for chromatographic methods [95,96,97,98]. These guidelines outline acceptance criteria for key analytical features, including accuracy, precision, specificity, detection, and quantification limits (LOD and LOQ), linearity, range, and robustness. Accuracy ensures results are close to the actual value, while precision assesses reproducibility. Specificity verifies the method’s ability to distinguish analytes from potential interferences. LOD and LOQ determine the lowest detectable and quantifiable concentrations, respectively. Linearity confirms the method’s response is proportional to analyte concentration within a defined range, and the range specifies the concentration interval over which the process is valid. Robustness evaluates the method’s reliability under varying conditions.

Selecting a suitable validation guideline based on specific interests, regional applicability, and current legislation. Adhering to these guidelines prevents confusion regarding analytical parameters and ensures consistency in method validation practices. Furthermore, documenting validation protocols and results is crucial for transparency and traceability, aligning with quality assurance and regulatory compliance standards established by organizations such as the FDA, EURACHEM, and ISO. This documentation enhances confidence in the reliability and accuracy of analytical data generated in food analysis. Despite the challenges, rigorous analytical method validation ensures reliable data for decision-making in research, quality control, and regulatory compliance efforts.

### 4.3. Novel Performance Metrics for Chromatographic Analytical Methods 

The increasing environmental awareness and regulatory demands have driven the development of metrics and tools to measure the “greenness” of analytical methods. Evaluating the sustainability of chromatographic methods involves considering numerous factors, including environmental impact, resource use, and safety considerations. Assessing the environmental impact of these methods can be particularly challenging, especially with complex sample preparation processes, which may include derivatization, multiple equipment usage, various procedural steps, solvent exchanges, and clean-ups. Recently, however, novel tools have been introduced to help researchers evaluate these impacts, standardize measurements, and facilitate comparisons across methods:The Analytical Eco-Scale evaluates environmental impact based on penalty points assigned to factors like reagent toxicity, energy consumption, and waste generation [99].The Analytical Greenness (AGREE) metric approach provides a quantitative score representing overall greenness, considering solvent usage, energy consumption, and waste [100].The Green Analytical Procedures Index (GAPI) visually represents the greenness of an analytical procedure across various stages using a color-coded hexagonal chart [101].Life Cycle Assessment (LCA) offers a holistic view of the environmental impacts throughout the method’s life cycle, from raw material extraction to disposal [102].

Additionally, the Blue Applicability Grade Index (BAGI) evaluates the practical applicability of analytical methods in various contexts, considering user-friendliness, cost-effectiveness, adaptability, scalability, and regulatory compliance [103]. Integrating these greenness metrics offers a comprehensive evaluation framework, ensuring that chromatographic methods are both environmentally sustainable and practical for diverse laboratory settings. This approach promotes sustainable and efficient analytical solutions, aligning with the increasing emphasis on environmental stewardship in scientific research. 

## 5. Conclusions

Chromatographic methods have significantly advanced the analysis of phenolic compounds in foods, each method offering unique practical applications and detection systems. HPLC with UV detection is widely used due to its robustness, established analytical protocols, and comprehensive spectrum libraries, which help identify phenolic structures. This technique involves solvent extraction, filtration, and separation through a high-pressure column, with phenolic compounds detected via UV light absorption at specific wavelengths. GC is another essential method for volatile phenolic compounds, distinguishing them based on boiling points and stationary phase interactions, with FID and MS providing high resolution and sensitivity. TLC and HPTLC are flexible, high-throughput methods that separate low molecular weight molecules on an adsorbent-coated plate, suitable for qualitative analysis via visual inspection or densitometry.

Recent advancements in chromatographic techniques have further enhanced food analysis capabilities. HILIC utilizes a polar stationary phase, improving retention and separation of hydrophilic compounds, and is compatible with various detection systems like MS, UV-Vis, DAD, and ELSD. Nano-LC aligns with green analytical chemistry principles by using smaller sample sizes and mobile phase volumes, enhancing resolution and detection limits. SFC employs supercritical CO₂ as the mobile phase, offering high efficiency, rapid separation, and reduced environmental impact, making it suitable for phenolic isomer separation with detection systems such as UV-vis and MS. MDC integrates multiple chromatographic methods for comprehensive separation of complex mixtures, improving resolution and peak capacity, and is particularly useful for profiling complex phenolic mixtures in foods. CE provides rapid, efficient separations with high resolution and sensitivity, ideal for bioactive compound analysis, and when combined with mass spectrometry (CE-MS), it offers detailed structural information and high-throughput capabilities. These advancements have overcome the limitations of conventional methods, enabling more precise, sensitive, and environmentally sustainable analysis of phenolic compounds in various food matrices and enhancing our understanding of their nutritional and functional properties.

Standardization and validation of chromatographic methods are essential for ensuring accuracy and precision in analyzing phenolic phytochemicals. Consistent protocols minimize systematic errors and enhance reproducibility, enabling reliable data comparisons across studies and laboratories. Adhering to validation guidelines provided by organizations like the FDA, EURACHEM, and ISO ensures the credibility of analytical results and meets regulatory requirements. However, challenges such as complex food matrices, achieving consistent precision and accuracy, and ensuring specificity and selectivity require ongoing research and rigorous validation practices.

Recent developments in metrics and tools to measure the “greenness” of analytical methods address the environmental impact of chromatographic techniques. Tools like the Analytical Eco-Scale, AGREEprep, GAPI, LCA, and BAGI help researchers evaluate environmental impacts, standardize measurements, and compare methods. Integrating these greenness metrics with conventional performance metrics provides a comprehensive evaluation framework, ensuring that chromatographic methods are both environmentally sustainable and practical for diverse laboratory settings. This approach promotes sustainable and efficient analytical solutions, aligning with the growing emphasis on environmental stewardship in scientific research. Despite the progress, further advancements are needed to fully realize the potential of these methods in food analysis.

## Figures and Tables

**Figure 1 foods-13-02268-f001:**
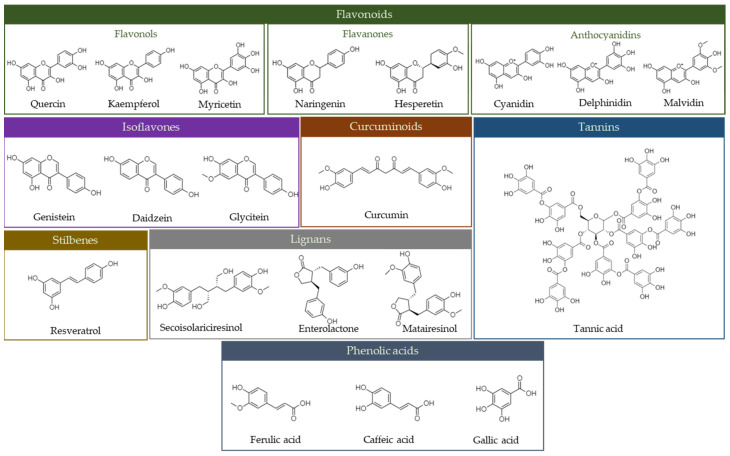
Classes of the most abundant phenolic phytochemicals (all molecules were drawn using ChemDraw 23.3 software).

**Figure 2 foods-13-02268-f002:**
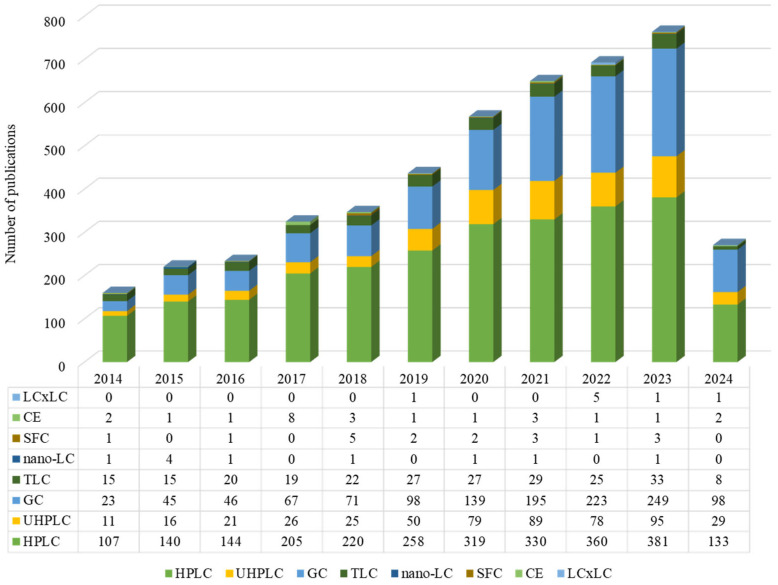
Trends in published research on chromatographic techniques for phenolic phytochemicals (2014–Present, according to the Web of Science). High-Performance Liquid Chromatography (HPLC), Ultra High-Performance Liquid Chromatography (UHPLC), Gas Chromatography (GC), Thin-Layer Chromatography (TLC), Nano-Liquid Chromatography (Nano-LC), Supercritical Fluid Chromatography (SFC), Capillary Electrophoresis (CE), and Two-Dimensional Liquid Chromatography (LCxLC).

**Figure 3 foods-13-02268-f003:**
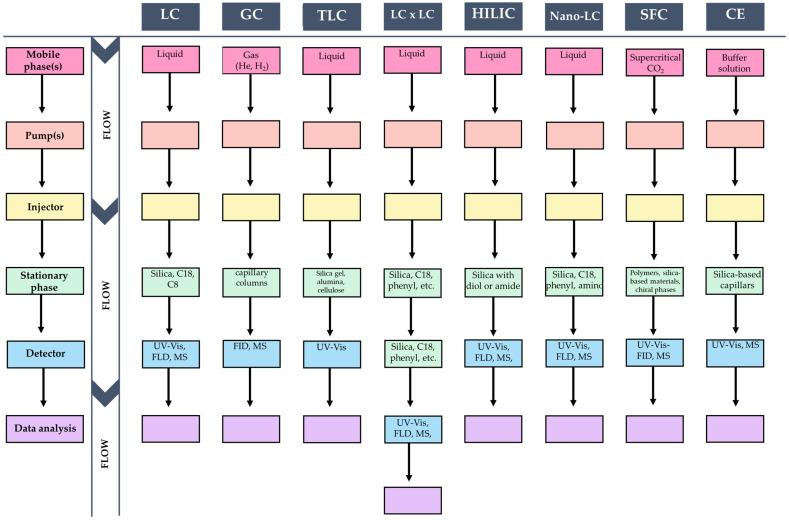
Comparative scheme of chromatographic techniques for phenolic phytochemical liquid chromatography (LC), including High-Performance Liquid Chromatography and Ultra High-Performance Liquid Chromatography, Gas Chromatography (GC), and Thin-Layer Chromatography (TLC), Nano-Liquid Chromatography (Nano-LC), Supercritical Fluid Chromatography (SFC), Capillary Electrophoresis (CE), and two-dimensional Liquid Chromatography (LCxLC).

**Table 1 foods-13-02268-t001:** Comparative overview of HILIC methods for phenolic phytochemical analysis of foods.

Sample	Analytes (*n*)	Stationary Phase	Mobile Phase	Flow (mL/min)	Detection System	Total Analysis Time (min)	Ref.
Yunnan large-leaf tea (*Camellia sentences*)	Amino acids (23) Alkaloids (9) Nucleosides (7) Nucleotides (6)	HILIC 1.7 μm column (100 × 2.1 mm)	0.2% FA and 5 mM AF in water (A) and ACN–water (90:10) (B)	0.4	MS/MS	8.5	[29]
Willow Bark (*Salix* sp.)	Phenolic compounds (2) Monosaccharides (2)	Luna^®^ Omega Sugar 3 µm column (250 × 4.6 mm)	ACN (A), water (B), MeOH (C)	1.3	DAD-ELSD	9.8	[30]
Red Garlic	Flavonols (3) Saccharides (3)	HILIC VG 50 4E 5 μm column (250 × 4.6 mm)	0,01% FA in ACN (A) and water (B)	0.6	DAD	40	[25]
Dietary supplements	B6 vitamin Catechins (4)	Column: eQuant™ ZIC-HILIC column 3.5 μm (100 × 2.1 mm)	8 mM FA (pH 2.8) (A), ACN (B)	0.2	MS	20	[31]
Green tea black tea chrysanthemum	Ningnanmycin	Poroshell 120 HILIC 1.9 μm column (150 × 2.1 mm)	ACN (A), 50 mM AF (B)	0.3	MS	15	[32]
Green, black, ginger, hibiscus, moringa, and fenugreek teas	Rutin	SEQuant ZIC-HILIC 3.5 μm column (100 × 4.6 mm)	35 mM NaAOc buffer (A), ACN (B)	0.5	UV-Vis	6	[33]
Apple juice	Phenolic acids (20)	Luna HILIC 3 μm column (150 × 2 mm)	0.1% AA in ACN (A), water/ACN/AA (79.9:2:0.1) (B)	0.3	DAD-HRMS	40	[34]
Hydrolyzed pomegranate peel.	Peptides 26	Ascentis Express 2.7 μm column (100 × 2.1 mm)	65 mM NaAOc in water (A), ACN (B)	0.3	Q-TOF	25	[35]
Vegetables	Amino acids (15) Vitamins b (7) Polyphenols (27)	HILIC-BEH-amide 1.7 μm column (2.1 × 100 mm)	0,02% FA in H20 (A), ACN (B)	0.2	MS/MS	20	[36]
Fresh and aged garlic	Phenolic compounds (10)	XBridge BEH-amide 3.5 μm column (150 × 4.6 mm	0,1% NH_4_OH in ACN (A) and water (B)	1.0	ELSD	35	[26]
Annurca and Red Delicious apple	Phenolic acids (6) Flavonoids (57)	Luna^®^ HILIC 3.0 μm column (150 × 2.0 mm)	AA/water/ACN (0,1:80:2) (A) 0,1% AA in ACN (B)	0.5	DAD-Q-TOF	25	[37]
Wild sea buckthorn berries	Catechins (11) Proanthocyanidin (49)	Luna HILIC 200 A 3 μm column (150 × 3.00 mm)	ACN (A), 0.5% FA in H20 (MPB)	0.6	DAD-MS	12	[38]
Blueberries red cabbage red radish grape skins black beans	Anthocyanins (6–19)	BEH Amide 1.7 μm column (150 mm × 1.0 mm)	0,4% TFA in ACN (A), in H20 (B)	0.2–1.0	DAD-Q-TOF	60	[39]
Peanut skins	Proanthocyanidins (60)	Princeton SPHER DIOL 5 μm column (250 × 4.6 mm)	2% AA in ACN (A), MeOH/water (95:3) (B)	1.0	MS	50	[40]
Moscato Rosa grapes	Phenolic acids (5) Flavonoids (5) Anthocyanins (3)	ZIC SeQuant 5 μm column (150 × 10 mm)	ACN (A), 1% FA (B)	0.2	DAD-ELSD-NMR-MS/MS	56	[41]

*n*, number of analytes; ref., references; FA, formic acids; AF, ammonium formate; ACN, acetonitrile; MeOH, methanol; NaAOc, sodium acetate; AA, acetic acid; TFA, trifluoro acetic acid; MS/MS: Tandem Mass Spectrometry; DAD: Diode Array Detector; ELSD: Evaporative Light Scattering Detector; MS: Mass Spectrometry; UV-Vis: UV-Visible Spectroscopy; Q-TOF: Quadrupole Time-of-Flight; HRMS: High-Resolution Mass Spectrometry; NMR, Nuclear Magnetic Resonance.

**Table 2 foods-13-02268-t002:** Nano-Liquid Chromatography methods for phenolic phytochemical analysis of foods.

Sample	Analytes (*n*)	Stationary Phase (particle Size, ID, Packed Length)	Mobile Phase	Flow (nL/min)	Detection System	Total Analysis Time (Min)	Ref.
Aloe plants	Anthrones (14)	ChromSpher 3 C18 (3 μm, 0.1 × 150 mm)	0.02 TFA water (A) and ACN (B)	350	UV-MS	32	[43]
Olive oil	Phenolic acids (1) Flavonoids (2) Other phenolic compounds (4)	BioSphere (3 μm, 0.075 × 100 mm)	0.5% AA in water (A), ACN (B)	300	MS	30	[44]
Tea	Phenolic acids (5) Flavonoids (6) Alkaloids (2) Caffeine	Kinetex C18 (2.6 μm, 0.1 × 100 mm)	0.5% FA in water (A) and ACN/MeOH (70:30 *v*/*v*) (B)	1200	UV-MS	15	[45]
Bee pollen	Phenolic acids (6) Hydroxycinnamic acids (5) Flavonoids (5)	Kinetex C18 (2.6 μm, 0.1 × 100 mm)	0.5% FA in water (A) and ACN (B)	500	UV-vis	20	[46]
Citrus juices	Flavonoids (7)	Hydride-based RP-C18 (2 μm, 0.075 × 100 mm)	1% FA in water (A) and ACN (B)	500	UV-vis	15	[47]
Dietary supplements	Flavonoids (5)	Hydride-based RP-C18 (2 μm, 0.075 × 100 mm)	0.55% FA in water (A) and ACN/MeOH (B)	450	UV-vis	5	[48]
Cranberry syrups	Phenolic acids (6) Flavonoids (35) Iridoids (6)	C18 (3 μm, 0.075 × 101 mm)	1% FA in water (A) and ACN (B)	300	Q-TOF-MS	40	[49]

*n*, number of analytes; ref., references; FA, formic acids; ACN, acetonitrile; MeOH, methanol; AA, acetic acid; TFA, trifluoro acetic acid; MS: Mass Spectrometry; UV-Vis: UV-Visible Spectroscopy; Q-TOF: Quadrupole Time-of-Flight.

**Table 3 foods-13-02268-t003:** Supercritical Fluid Chromatography methods for phenolic phytochemical analysis of foods.

Sample	Analytes (*n*)	Stationary Phase (Particle Size, Id, Packed Length)	Mobile Phase	Pressure MPa	Flow (mL/min)	Detection System	Total Analysis Time (Min)	Ref.
Radix Hedysari	Phenolic acid (1) Flavonoid (7) Other phenolic compounds (3)	HSS SB C18 1.7 μm column (150 × 2.1 mm)	CO_2_ (A) and 0.2% FA in MeOH (B)	11.03	1.5	DAD	25	[52]
Wild ivy	Chlorophyll and derivates (31)	Cortecs C18 2.7 μm column (150 × 4.5 mm)	CO_2_/MeOH (80:20 *v*/*v*)	10	1.5	MS	10	[53]
Bee pollen	Flavanones (10)	Chiralpak AD 10 μm column (250 × 4.6 mm)	CO_2_ and EtOH/MeOH (80:15 *v*/*v*)	15	3	PDA	40	[54]
Alpinia officinarum	Flavonoids (3) Other phenolic compounds (7)	Diol 5 μm column (250 × 4.6 mm)	CO_2_ and MeOH (5–20%)	13.8	3	UV-vis	30	[55]
Walnut oil	Terpenoids (9)	BEH-2EP 1.7 μm column (100 × 3 mm)	0.1% FA in CO_2_ and MeOH	17	0.1	TQ-MS	13	[56]
Eucalyptus	Terpenes (17)	Super Carbon LC 2.7 μm column (150 × 3 mm)	CO_2_ and MeOH (5–20%)	22.75	1.5	MS/MS	7.5	[57]

*n*, number of analytes; ref., references; FA, formic acids; MeOH, methanol; EtOH, ethanol; DAD, Diode Array Detector; MS: Mass Spectrometry; PDA, PhotoDiode Array detector; UV-Vis: UV-Visible Spectroscopy; MS/MS, Tandem Mass Spectrometry.

**Table 4 foods-13-02268-t004:** Analytical methods for phenolic phytochemical determination in foods using comprehensive Two-Dimensional Liquid Chromatography.

Sample	Analytes (*n*)	Stationary Phase	Mobile Phase	Flow (mL/min)	Detection System	Total Analysis Time (min)	Ref.
Safflower	Flavonoids (75) Alkaloids (10)	XBridge Amide 3.5 μm column (150 × 4.6 mm)	0.1% FA in water (A) and ACN (B)	0.08	DAD-HRMS	170	[62]
		Ultimate amide five μm column (50 × 4.6 mm)	0.2 uM AF in water (A), ACN (B)	3			
Rhus coriaria	Phenolic acids (83)	SEQuant ZIC-HILIC 3.5 μm column (150 × 1.0 mm) *	0.1 % FA in water, pH 3.0 (A) and ACN (B)	0.01	PDA-MS	60	[63]
		Ascentis Express C18 2.7 μm column (50 × 4.6 mm)	0.1 % FA in water, pH 3.0 (A) and ACN (B)	3			
Cannabis	Cannabinoids (41) Procyanidins (6) Phenolic acids (4) Flavonoids (11)	Kinetex PFP 1.7 μm column (150 × 2.1 mm)	0.1% FA in water (A) and MeOH (B)	0.05	DAD-Q-TOF-MS	65	[64]
		Kinetex C18 2.6 μm column (50 × 4.6 mm)	0.1% FA in water (A) and ACN (B)	2.5			
Gelsemium elegans	Alkaloids (256)	XCharge C18 3.0 μm column (150 × 2.1 mm)	0.1% FA in water (A) and MeOH (B)	0.04	Q-TOF-MS	120	[65]
		BEH Shield C18 1.7 μm column (50 × 3 mm)	0.125% NH_4_OH in water (A) and MeOH (B)	1			
Cuscuta Chinensis	Organic acids (26) Flavonoids (45) Lignans (45) Phenolic acids (40) Alkaloids (5)	XBridge Amide 3.5 μm column (150 × 4.6 mm)	0.1% FA in water (A) and ACN (B)	1	UV-Q-TOF-MS	53	[66]
		Zorbax SB-AQ 1.8 μm column (100 × 2.1 mm)	0.1% FA in water (A) and ACN (B)	0.3			
Ceylon tea	Catechins and derivates (31)	Poroshell HPH-C18 2.7 μm column (150 × 2.1 mm)	0.1% FA in water (A) and MeOH (B)	0.12	HRMS	50	[67]
		Poroshell Bonus RP1.9 μm column (50 × 3.0 mm)	0.1% FA in water (A) and MeOH (B)	0.86			
Cannabis sativa	Cannabinoids (10) Terpenes (15)	Zorbax SB-CN 5 μm column (250 × 4.6 mm)	0.05% FA in MeOH/water (A) and ACN/water (B)	0.7	DAD	75	[68]
		Poroshell 120-SB 2.7 μm column (50 × 2.1 mm)	0.05% FA in water (A) and ACN (B)	2.7			
Bilberry, blackcurrant, blueberry, chokeberry, elderberry, honeyberry, and raspberry	Phenolic compounds (80)	Ascentis Express C18 2.7 μm column (50 × 4.6 mm)	0.1% FA in water (A) and ACN (B)	0.1	PDA-MS	80	[69]
		SEQuant ZIC-HILIC 3.5 μm column (150 × 1.0 mm)	0.1% FA in water (A) and ACN (B)	1			
Grape seeds, Rooibos tea, Wine, and Grapes	Phenolic compounds (156)	Xbridge Amide 1.7 μm column (150 × 1.0 mm)	0.1% FA in water (A) and ACN (B)	0.5	DAD-HRMS	70	[70]
		Kinetex C 18 1.7 μm column (50 × 3.0 mm)	0.1% FA in water (A) and ACN (B)	2.5			
Green tea	Anthocyanins (19)	Nomura Chemical Develosil Diol-100 5 μm column (250 × 1 mm)	10% FA in ACN (A) and H20 (B)	0.2	DAD-Q-TOF-MS	60	[71]
		Zorbax SB-C18 1.8 μm column (50 × 4.6 mm)	0.4% TFA in ACN (A) and H20 (B)	1			
Apples	Phenolic compounds (65)	Lichrospher diol-5 5 μm column (150 × 1.0 mm)	2% AA in ACN (A) MeOH/water/AA acid (95:32) (B)	0.9	MS	50	[72]
		Ascentis Express C18 2.7 μm column (50 × 4.6 mm)	0,1% FA in water (A) ACN (C)	3			
Uncaria sessilifructus ^a^	Alkaloids (85) Phenolic acids (29)	PhenoSphere TM SCX 5 μm column (250 × 4.6 mm) *	20 mM AA/0.05% FA in water (A) and MeOH (B)	1	HRMS	70	[73]
		Acchrom XAmide 5 μm column (150 × 4.6 mm) *	0.1% FA in water (A) and ACN (B)	0.8			
		CSH Phenyl-Hexyl 1.7 μm column (100 × 2.1 mm)	0.1% FA in water (A) and ACN (B)	0.3			

*n*, number of analytes; ref., references; FA, formic acids; ACN, acetonitrile; MeOH, methanol; EtOH, ethanol; DAD, Diode Array Detector; MS: Mass Spectrometry; PDA, PhotoDiode Array Detector; Q-TOF: Quadrupole Time-of-Flight; HRMS: High-Resolution Mass Spectrometry; ^a^ Three-Dimensional Liquid Chromatography, * HILIC column.

**Table 5 foods-13-02268-t005:** Capillary Electrophoresis (CE) methods for phenolic phytochemical determination in foods.

Sample	Analytes (*n*)	Capillary (Length × I.D.)	Separation Medium	Separation Mechanism	Detection System	Total Analysis Time (min)	Ref.
Sunflower honey	Phenolic acids (11) Flavonoids (3) Other phenolic acids (1)	Fused silica (90 cm × 50 µm)	0.5 M NH_4_OH	CZE	MS	15	[74]
Chamomile flowers	Flavonoids (2)	Quartz (75 cm × 50 µm)	0.026 M borax	CZE	UV-vis	25	[75]
Salvia	Phenolic acids (2)	Fused silica (67 cm × 50 µm)	0.020 M borax	CZE	UV-vis	10	[76]
Shihu ^a^	Phenols (11)	Electropak™ C18 column (20 cm × 100 µm)	0.012 M borax in ACN	pCEC	UV-vis	35	[77]
Carob pekmez	Phenolic acids (10) Flavonoids (4) Phenolic aldehyde (1)	Fused silica (55 cm × 50 µm)	0.04 M borax	CZE	DAD	20	[78]
Soybean	Metabolites (198)	Fused silica (100 cm × 50 µm) ^b^ COSMO (+) (100 cm × 50 µm) ^c^	1M FA ^b^ 0.05 M AA ^c^	CZE	MS	40	[80]
Cranberries, cranberry juice, blueberries, grapes, grape juice, and raisins	Proanthocyanidins (4)	Fused silica (75 cm × 50 µm)	0.035 M borax in 5% MeOH	CZE	UV	40	[81]

*n*, number of analytes; ref. references; FA, formic acids; ACN, acetonitrile; MeOH, methanol; DAD, Diode Array Detector; MS: Mass Spectrometry; UV-Vis: UV-Visible Spectroscopy; capillary zone electrophoresis; pCEC, pressurized capillary electrochromatography; AA, ammonium acetate; ^a^ conventional Chinese medicine plant; ^b^ anionic metabolites; ^c^ cationic metabolites.

## Data Availability

No new data were created or analyzed in this study. Data sharing is not applicable to this article.

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
