# Peer review of "Advances in Chromatographic Analysis of Phenolic Phytochemicals in Foods: Bridging Gaps and Exploring New Horizons"

_foods, 2024, doi:10.3390/foods13142268_

Round 1
Reviewer 1 Report
Comments and Suggestions for Authors
The manuscript "Advances in chromatographic analysis of phenolic phytochemicals in foods: bridging gaps and exploring new horizons" aims to review recent advancements in chromatographic techniques for analysing phenolic phytochemicals in foods, focusing on both traditional and advanced methods. It discusses the benefits, limitations, and innovations in chromatographic methods and highlights the need for standardised procedures to ensure reliable and reproducible results. The manuscript is in the aim and scope of the Foods journal; however, there is no order of presentation. Furthermore, the manuscript lacks references. I suggest "rejection" in the present form because it requires too much improvement to be published.
The title should not be in all caps. Please correct this formatting issue. Check abbreviation.
Ensure that abbreviations are used consistently and not repeated unnecessarily. For example, UHPLC is mentioned in both L13-14 and L36. Avoid repeating abbreviations after they have been defined once.
Maintain consistency in the use of grammar and capitalisation. For instance, in paragraph 2, "Chromatographic," "Techniques," and "Phenolic" are capitalised without clear reasoning. Ensure uniformity throughout the manuscript.
The manuscript would benefit from more references to support various statements:
L27-L29: Add references for the statement "Phenolic...health."
L31-L33: Add references for "The meticulous...impacts."
L35-L37: Add references for "Techniques...profiles."
L103-L108, L119-L116, L145-L155, L157-L161, L177-L186, L187-L194, L212-L214, L217-L228, L250-L255, L272-276, L287-L299, L300-L309, L312-L323, L351-L358, L367-L397, L413-L430: Include relevant references in these sections.
L61-L64: "Phenolic…color". It can be added to the introduction part.
The manuscript currently lacks a clear structure. The section on "Traditional Chromatographic Techniques for Phenolic Phytochemicals" should define what is meant by "traditional methods" and how these are distinguished from "advancements in liquid chromatography-based methods."
Consider restructuring to discuss chromatography techniques separately from detectors. First, describe the chromatographic parts, outlining their advantages and disadvantages, and then discuss the associated detectors in a similar manner.
Create a Figure 2 that includes a search for detectors, such as "Nano-LC" or "SFC." This would provide a more comprehensive analysis of advanced methods.
Explain how Figure 1 was created and ensure it includes methods like HILIC, CEC, CE, MDC, and LCxLC.
Ensure the caption for "Table 2" is not repeated.
Relying solely on Web of Science (WoS) for the review has limitations. WoS may not cover all relevant scientific publications, and significant articles might be found in other databases such as Scopus or PubMed.
Be consistent in describing analytical techniques. For instance, HPLC-UV is discussed in L70, and hyphenated techniques are mentioned in L148.
For example, lines L69-81, "chromatographic…periods", can be applied to all the chromatography analyses while you speak about HPLC-UV. Also, HPLC-MS/MS has a "the procedure involves..column". The same is true with "The operating mechanism…purity". The same as "Under high pressure…material". In this way, you can't distinguish properly between analytical techniques. For these reasons, I strongly suggest speaking about the chromatography parts and then about each associated detector.
Describe chromatography and detectors separately to provide a clearer distinction between different techniques. Describe the advantages and disadvantages for each detector and for each chromatography part.
The section on validation guidelines for chromatographic methods (L435-L455) seems out of scope. Consider omitting or relocating this section.
Author Response
Please see the attachment.
The authors wants to thank the reviewer for their valuable comments that helped improve our manuscript.

Reviewer 2 Report
Comments and Suggestions for Authors
This article is interesting, well written and organized, however some observations and comments are made.
I consider that a table or figure could be attached with the classification of phenolic compounds and their general structures to give an idea of ​​the different compounds that may be more related to chromatographic techniques.
Figure 1. I recommend that the years on the X axis be in increasing order.
Lines 230.231. Please check this phrase “SFC is an innovative technique that utilizes supercritical fluids, typically COâ‚‚, as the mobile phase. COâ‚‚.”
What factors can affect or improve the resolution of different chromatographic techniques? How does temperature, pH, etc. influence, about the separation of phenolic compounds in each method?
It is suggested to place a diagram of the operation of each technique so that the difference between chromatographic techniques is better understood.
What is new about the information provided in this review if there are already articles published on this same topic? What differentiates this review from other existing ones?
Author Response
Please see the attachment.
We would like to thank the reviewer for their valuable comments.

Round 2
Reviewer 1 Report
Comments and Suggestions for Authors
The manuscript has been improved a lot. Therefore, I recommend the publication.
Reviewer 2 Report
Comments and Suggestions for Authors
The authors increased the quality of the article by responding to the suggestions.